# PPP and PPP-AR Kinematic Post-Processed Performance of GPS-Only, Galileo-Only and Multi-GNSS

**Georgia Katsigianni** [1,2,*] ⓘ**, Sylvain Loyer** [2] **and Felix Perosanz** [1]

[1] Centre National d'Etudes Spatiales (CNES), Observatoire Midi-Pyrénées (OMP), Géosciences Environnement Toulouse (GET), 14 avenue Edouard Belin, 31400 Toulouse, France; felix.perosanz@cnes.fr

[2] Collecte Localisation Satellites (CLS Group), 8-10 rue Hermès, 31520 Ramonville Saint Agne, France; sloyer@groupcls.com

* Correspondence: georgia.katsigianni@cnes.fr; Tel.: +33-(0)5-6133-2892

**Abstract:** Precise point positioning (PPP) has been used for decades not only for general positioning needs but also for geodetic and other scientific applications. The CNES-CLS Analysis Centre (AC) of the International GNSS Service (IGS) is performing PPP with phase ambiguity resolution (PPP-AR) using the zero-difference ambiguity fixing approach also known as "Integer PPP" (IPPP). In this paper we examine the postprocessed kinematic PPP and PPP-AR using Galileo-only, GPS-only and Multi-GNSS (GPS + Galileo) constellations. The interest is to examine the accuracy for each GNSS system individually but also of their combination to measure the current benefits of using Galileo within a Multi-GNSS PPP and PPP-AR. Results show that Galileo-only positioning is nearly at the same level as GPS-only; around 2–4 mm horizontal and aound 10 mm vertical repeatability (example station of BRUX). In addition, the use of Galileo system—even uncompleted—improves the performance of the positioning when combined with GPS giving mm level repeatability (improvement of around 30% in East, North and Up components). Repeatabilities observed for Multi-GNSS (GPS + GAL) PPP-AR, taking into account the global network statistics, are a little larger, with 8 mm in horizontal and 17 mm in vertical directions. This result shows that including Galileo ameliorates the best positioning accuracy achieved until today with GPS PPP-AR.

**Keywords:** GNSS; GPS; Galileo; precise point positioning (PPP); PPP with phase ambiguity resolution (PPP-AR); integer PPP (IPPP); ambiguity resolution; zero-difference ambiguity fixing; multi-GNSS

## 1. Introduction

The precise point positioning method (PPP) [1] is a well-known and widely used method for positioning using zero-difference observations. This method is used for calculating the coordinates of a station without the need of a reference station nearby as a control station. It has been used in numerous scientific applications; namely general static positioning [2] for local or global networks, kinematic positioning [3], time transfer [4], etc. The PPP accuracy reaches the highest levels once the carrier phase ambiguities are resolved—the so-called PPP-AR (PPP with ambiguity resolution) or less frequently integer PPP (IPPP). The French Space Agency, Centre National d'Etudes Spatiales (CNES) and the Collecte Localisation Satellites company (CLS Group) Analysis Centre (AC) of the International GNSS Service (IGS) use an undifferenced ambiguity resolution solution for GPS [5] and Galileo [6]. Such a method can be applied also for the determination of precise orbits and integer recovery clocks (IRC). In order to apply this method for PPP and PPP-AR for GPS and Galileo, satellite orbits and IRC products as well as satellite biases have to be consistent. In our study they are taken

from the CNES-CLS AC. There are two conditions for this method: for PPP or PPP-AR, orbit and clock products have to be consistent and of the best quality; for PPP-AR IRCs and associated methods, biases need to be available for the products used. For the first condition, any precise orbit like the ones made available by the International GNSS Service (IGS) or Analysis Centers (ACs) in the frame of the Multi GNSS pilot project (MGEX) [7] can be used. For the second we can use the products delivered routinely by the CNES-CLS AC for GPS [8] and Galileo [6].

So far there are already publications giving the possibilities of Galileo positioning [9] or studying the performance the benefit of using multi-GNSS measurements in PPP [7,8]. In a previous study [6] we presented the zero-differences approach used to compute IRC products for Galileo and GPS. Recently we presented the positioning capabilities of Galileo-only solutions [10]. In this article we examine the performance and precision of PPP and PPP-AR in a comparison of Galileo-only, GPS-only and GPS + Galileo AR solutions.

This study is done using the GPS and the incomplete Galileo constellation. As of 2019, the Galileo constellation comprises 26 satellites in total (4 In-Orbit Validation (IOV) and 22 Full Operational Capability (FOC)). Two of the FOC satellites are in eccentric orbits, one IOV is not available and one FOC is not usable. The total number of usable satellites is 22: 3 IOV and 21 FOC satellites [11]. The full constellation is only 4 satellites away and it is scheduled for 2020 [12]. The following hypothesis is examined and verified: if GPS and Galileo measurements are of the same quality, then the simultaneous use of the two systems will give better accuracy than when using each system separately. If the two systems are compatible, then the number of measurements is nearly double than in a single-system solution, and the number of parameters (apart from biases for the different frequencies used) is the same.

This publication is organized in the following sections. Firstly, the undifferenced ambiguity resolution and PPP-AR processing is briefly presented. Section 3 is devoted to the experiments, the processing and the results. Finally, in Section 4, some conclusions are given together with suggestions for further work and perspectives.

## 2. Materials and Methods

The PPP with ambiguity resolution processing has the advantage that the processing can be done directly at the user level without any reference station around. The requirement, however, is that precise and consistent satellite products (derived from a global network of stations) must be available. In this study products are taken from the CNES-CLS AC: satellite orbit file (.sp3), clock file (.clk) and the Wide-Lane satellite biases (.wsb) (given either in the header of the clock files or taken from the CNES/CLS portal. Files Wide_lane_GAL_satellite_biais.wsb and Wide_lane_GPS_satellite_biais.wsb are available at ftp://ftpsedr.cls.fr/pub/igsac/).

The used zero-difference ambiguity fixing method [8,13] equation model bellow is given for the pseudorange (code) and carrier phase measurements for two frequencies:

$$P^s_{r,i} = \rho^s_r + c\Delta t + T^s_r + I^s_r + b^s_i + b_{r,i} + E^s_{r,i} \tag{1}$$

$$P^s_{r,j} = \rho^s_r + c\Delta t + T^s_r + \frac{f^2_i}{f^2_j}I^s_r + b^s_j + b_{r,j} + E^s_{r,j} \tag{2}$$

$$L^s_{r,i} = \lambda_i \varphi^s_{r,i} = \rho^s_r + c\Delta t + T^s_r - I^s_r + \lambda_i N^s_{r,i} + \lambda_i W^s_r + \beta^s_i + \beta_{r,i} + \varepsilon^s_{r,i} \tag{3}$$

$$L^s_{r,j} = \lambda_j \varphi^s_{r,j} = \rho^s_r + c\Delta t + T^s_r - \frac{f^2_i}{f^2_j}I^s_r + \lambda_j N^s_{r,j} + \lambda_j W^s_r + \beta^s_j + \beta_{r,j} + \varepsilon^s_{r,j} \tag{4}$$

where:

- $P_{r,i}^s$, $P_{r,j}^s$ are the code measurement at receiver $r$ from satellite $s$ on frequency $i$ or $j$ (m)
- $L_{r,i}^s$, $L_{r,j}^s$ are the phase measurement at receiver $r$ from satellite $s$ on frequency $i$ or $j$ (m)
- $\rho_r^s$ is the geometric distance between receiver and satellite (m)
- $\Delta t$ ($\Delta t = \delta t_r - \delta t^s$) is the clock correction related to the satellite ($\delta t^s$) and the receiver ($\delta t_r$) with respect to the synchronization to the GPS time (s)
- $T_r^s$ is the troposphere delay (m)
- $I_r^s$ is the ionosphere delay (m)
- $E_{r,i}^s$, $E_{r,j}^s$ are the code measurement errors at receiver $r$ from satellite $s$ on frequency $i$ or $j$ (m) including all sources of code errors: multipath and noise.
- $f_i$, $f_j$ are the carrier frequency $i$ or $j$ (Hz)
- $c$ is the speed of light in vacuum (m/s)
- $\lambda_i$, $\lambda_j$ are the nominal wavelength of the carrier frequency $i$ or $j$ (m)
- $\varphi_{r,i}^s$, $\varphi_{r,j}^s$ are the carrier phase measurement at receiver $r$ from satellite $s$ on frequency $i$ or $j$ (cycles)
- $N_{r,i}^s$, $N_{r,j}^s$ are the integer carrier phase ambiguity at receiver $r$ from satellite $s$ on frequency $i$ or $j$
- $W_r^s$ is the carrier phase wind up effect (cycles)
- $b^s$, $b_r$ are the code phase biases of satellite and receiver (m)
- $\beta^s$, $\beta_r$ are the carrier phase biases of satellite and receiver (m)
- $\varepsilon_{r,i}^s$, $\varepsilon_{r,j}^s$ are the carrier phase measurement error at receiver $r$ from satellite $s$ on frequency $i$ or $j$ (m) including all sources of phase errors, remaining uncorrected phase center offset and phase center variation, multipath and noise.

In this study the frequencies used for the GPS system are in the band of $L1$ and $L2$ ($f_{L1} = 154 \times f_0$ and $f_{L2} = 120 \times f_0$). For the Galileo system are in the band of $E1$ and $E5a$ ($f_{E1} = 154 \times f_0$ and $f_{E5a} = 115 \times f_0$), where $f_0 = 10.23$ MHz.

From all these four equations it is possible to form a Melbourne–Wübbena linear combination that has the identity to reduce measurement noise and to cancel out any geometric, ionospheric and clock terms [14,15]:

$$
\begin{aligned}
MW_r^s &= \lambda_{wl}\varphi_r^s = \left(\frac{f_i}{f_i-f_j}L_{r,i}^s - \frac{f_j}{f_i-f_j}L_{r,j}^s\right) - \left(\frac{f_i}{f_i+f_j}P_{r,i}^s + \frac{f_j}{f_i+f_j}P_{r,j}^s\right) \\
&= \lambda_{wl}(N_{r,i}^s - N_{r,j}^s - \mu^s + \mu_r(t)) = \lambda_{wl}\left(N_{wl,r}^s - \mu^s + \mu_r(t)\right)
\end{aligned}
\tag{5}
$$

where:

- $MW_r^s$ is the Melbourne-Wübbena linear combination at receiver $r$ from satellite $s$ (m)
- $\lambda_{wl}$ ($\lambda_{wl} = c/(f_i - f_j) = \lambda_i\lambda_j/(\lambda_j - \lambda_i)$) is the wide-lane (WL) wavelength (m)
- $N_{wl,r}^s$ ($N_{wl,r}^s = N_{r,i}^s - N_{r,j}^s$) is the WL ambiguity at receiver $r$ from satellite $s$
- $\mu^s$ is the delay coming from the satellite (also known in the bibliography as WL satellite bias (WSB))
- $\mu_r(t)$ is the delay coming from the receiver (also known in the bibliography as WL receiver bias (WRB))

It has been observed that for the GPS system the $\mu^s$ are stable over long periods of time and can be considered as constant during at least one day [16]. For the Galileo system they are stable for longer periods; up to months [10]. The $\mu_r(t)$ delay is considered to vary over time because it depends on the behavior of each receiver.

As it is seen from the Equation (5), the terms $N_{wl,r}^s$, $\mu^s$ and $\mu_r(t)$ are totally correlated. Normally, the $\mu^s$ is known; i.e., it is calculated and provided from the CNES/CLS AC to the users [6,8]. The $\mu_r$ (one per epoch of measurements) and the $N_{wl,r}^s$ (one per satellite pass) can be separated and solved

using all available equations (Equation (5) corrected by the $\mu^s$) from all satellites in view, using a Least Squares Estimation (LSE) processing associated with a bootstrap method [17,18]. The result after this step are the $N_{wl,r}^s$ and $\mu_r(t)$.

Once the $N_{wl,r}^s$ is determined, the following step is to form an ionosphere-free linear combination that has the property to cancel out the first order of ionospheric effects. These combinations use the equations in the two frequencies where they apply to them a coefficient: $\alpha_i$ to the frequency $i$ and $\alpha_j$ to the frequency $j$ respectively [19].

$$\alpha_i = \frac{f_i^2}{f_i^2 - f_j^2} \tag{6}$$

$$\alpha_j = \frac{-f_i^2}{f_i^2 - f_j^2} \tag{7}$$

Using the above coefficients and Equations (6) and (7), the ionosphere-free linear combinations for code and carrier phase become:

$$P_{r,IF}^s = \alpha_i P_{r,i}^s + \alpha_j P_{r,j}^s = \rho_r^s + c\Delta t + T_r^s + E_{r,IF}^s \tag{8}$$

$$
\begin{aligned}
L_{r,IF}^s &= \alpha_i L_{r,i}^s + \alpha_j L_{r,j}^s \\
&= \rho_r^s + c\Delta t + T_r^s + \frac{c}{f_i^2 - f_j^2}\left(f_i N_{r,i}^s - f_j N_{r,j}^s\right) + \frac{c}{f_i^2 - f_j^2}\left(f_i W_r^s - f_j W_r^s\right) + \varepsilon_{r,IF}^s \\
&= \rho_r^s + c\Delta t + T_r^s + \lambda_{nl} N_{r,i}^s + \frac{\lambda_{nl}\lambda_{wl}}{\lambda_j} N_{wl,r}^s + \lambda_{nl} W_r^s + \varepsilon_{r,IF}^s
\end{aligned}
\tag{9}
$$

where:

- $P_{r,IF}^s$ is the ionosphere-free code measurement at receiver $r$ from satellite $s$ (m)
- $E_{r,IF}^s$ is the ionosphere-free code measurement error at receiver $r$ from satellite $s$ (m)
- $L_{r,IF}^s$ is the ionosphere-free carrier measurement at receiver $r$ from satellite $s$ (m)
- $\varepsilon_{r,IF}^s$ is the ionosphere-free carrier measurement error at receiver $r$ from satellite $s$ (m)
- $\lambda_{nl}$ $\left(\lambda_{nl} = c/\left(f_i + f_j\right) = \lambda_i\lambda_j/\left(\lambda_i + \lambda_j\right)\right)$ is the narrow-lane (NL) wavelength (m)

The following Table 1 gives the wide-lane and the narrow-lane wavelengths for GPS and Galileo frequencies used in this article.

**Table 1.** Values for wide-lane and narrow-lane wavelength.

| GNSS Frequency | $\lambda_{wl}$(m) | $\lambda_{nl}$(m) |
|---|---|---|
| GPS (*L1*, *L2*) | 0.862 | 0.107 |
| Galileo (*E1*, *E5a*) | 0.751 | 0.109 |

The term $W_r^s$ is calculated by using the models proposed by Kouba [20]. All equations are then gathered to form another LSE processing. The system of equations for the ionosphere-free code and phase measurements are modelled according to the models given in Table 2 and used to estimate all the parameters: stations position, tropospheric parameters, receiver clocks and inter system bias in case of multi-GNSS measurements. For the PPP case the $N_{r,i}^s$ are solved with the other parameters as real values. For the PPP-AR case they are fixed to integer numbers within a bootstrap processing [8,13]. Once the integer $N_{r,i}^s$ is determined, the final PPP-AR solution is performed.

**Table 2.** Parameters, models and strategy of experiments.

| | |
|---|---|
| **Processing Strategy** | |
| Software | GINS, DYNAMO, EXE_PPP [21] |
| Strategy | PPP, PPP-AR zero-difference |
| Estimation | Static with Kalman, 300 s sampling |
| **Orbit, Clocks and Satellites Biaises** | |
| Orbits and clocks | CNES-CLS orbits ('grg') [8] |
| Satellite biases | CNES-CLS wide-lane satellite biases [22] |
| GNSS relative weighting | Equal weighting for GPS and Galileo |
| GNSS measurement sigmas (at 0° of elevation) | Code: 60 cm; Phase: 3.5 mm |
| Elevation cut-off | 8 deg |
| Elevation weighting function, where $\vartheta$ is the elevation angle | $\sigma(\vartheta) = \frac{0.0035}{0.15 + 0.85 \sin \vartheta}$ |
| **Models for Processing** | |
| Antenna phase center corrections | ANTEX14 PCO/PCV [23] |
| Troposphere model | VMF1 [24] + GPT2 [25] (A priori local meteorological parameters (pressure, temperature, and wet mapping function coefficients) of GPT2 model are used to compute hydrostatic delays and for the wet mapping function VMF1. We then adjust 1 zenithal tropospheric delay per two hours in factor of the wet mapping function). |
| Ionosphere model | Ionosphere-free combination and second ordercorrections [26] |
| Reference frame | ITRF 2014 [27] |
| Attitude model | Kouba [28] for GPS and GSA [23] for Galileo |
| Ocean loading effects | FES2012 [29] |
| Earth orientation modelling | IERS Conventions 2010 [30] |
| Earth orientation parameters | EOP C04 [31] |
| Phase windup | Models used by Kouba [28] |
| **Estimated Parameters** | |
| Troposphere | 1 ZTD/2 h, 1 pair of gradients (E, N)/day (1 couple of gradients in north and east direction are also adjusted per day following [31,32] |
| Observation sampling | 300 s |
| Inter-system biases (phase obs.) | 1 per station (zero mean condition) |
| Station coordinates estimates | X, Y, Z transformed to East, North, Up |

Figure 1 is giving a graphic overview of the PPP and PPP-AR method used. In the beginning, the satellite $\mu^s$ are needed together with the station RINEX file. The pre-processing phase uses the $MW_r^s$ observations formed from the individual RINEX observations and the GPS and Galileo biases $\mu^s$ to solve the $N_{wl,r}^s$ for integers. Then the $N_{wl,r}^s$ and satellite orbit and clock products are used as inputs to form the ionosphere-free combination measurements (code and carrier phase) for the first processing. The system of equations is solved for the PPP solution. In the case of PPP-AR processing the proceedure continues further after having fixed the $N_{r,i}^s$ to integers; in this second processing to give the PPP-AR solution mostly ionosphere-free ambiguity-free carrier phase measurements are used for GPS and/or Galileo.

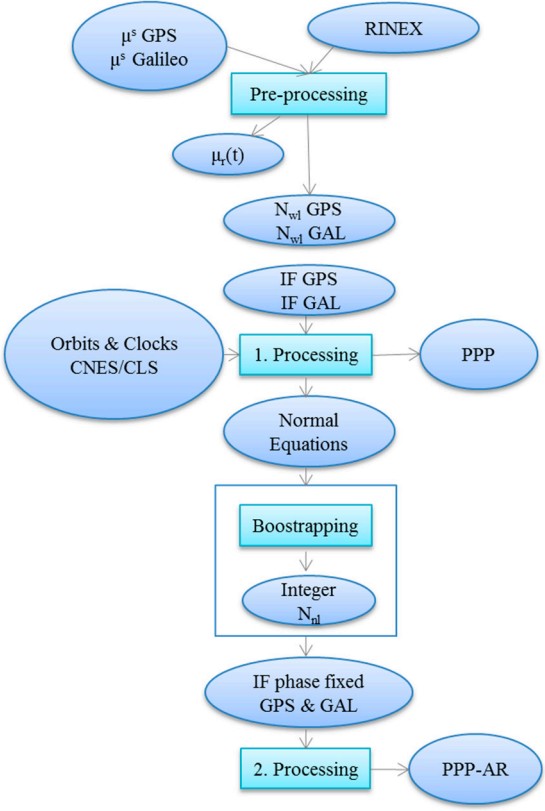

**Figure 1.** Steps of the procedure to perform precise point positioning (PPP) and PPP with phase ambiguity resolution (PPP-AR) of a combined Multi-GNSS solution (GPS + Galileo).

For the experimentation, one week of data (11–17 February 2019, Day of Year (DOY): 42–49/2019) is chosen. During that period, 31 GPS satellites and 24 Galileo satellites (including the ecliptic E14 and E18) were processed. All models and processing parameters and given in Table 2.

The IGS compiles a consistent set of absolute Antenna Phase Centre (APC) (i.e., Phase Center Offset (PCO) and Phase Center Variation (PCV)) corrections for both ground stations and satellites antennas, which are provided in so-called the Antenna Exchange Format (ANTEX) files [20]. These are very important for the calculation of the geometrical distance between satellite and receiver. For the APC corrections, it was decided to use the ANTEX14 file from the IGS. During the processing of the present article, ANTEX14 file included the APC values of the receivers for GPS L1/L2 frequencies, but not of Galileo E1/E5a frequencies. Delivery of the respective receiver APC values for the Galileo E1/E5a frequencies was underway from the IGS Antenna WG [33].

For the ambiguity fixing step biases, we do not use any additional bias for Galileo ambiguity fixing since a previous study [6] showed that the Galileo $\mu^s$ biases were compatible with all kind of receivers and modulations (such as L1C, L1W, L1X, etc.).

## 3. Results

The following section shows some examples of PPP and PPP-AR positioning in detail as well as global network summary graphs in East (E), North (N) and Up (U) components.

### 3.1. Some Station Examples

The station BRUX from the IGS network has been chosen to show in detail the temporal series for the entire week, as examples of good PPP and PPP-AR repeatability. Figure 2 gives examples for PPP and PPP-AR solutions for Galileo-only, GPS-only and their combinations. For each temporal series the

mean value is calculated and subtracted to center the series to zero. In addition, Table 3 shows the percentages of ambiguity resolution for the PPP-AR mode.

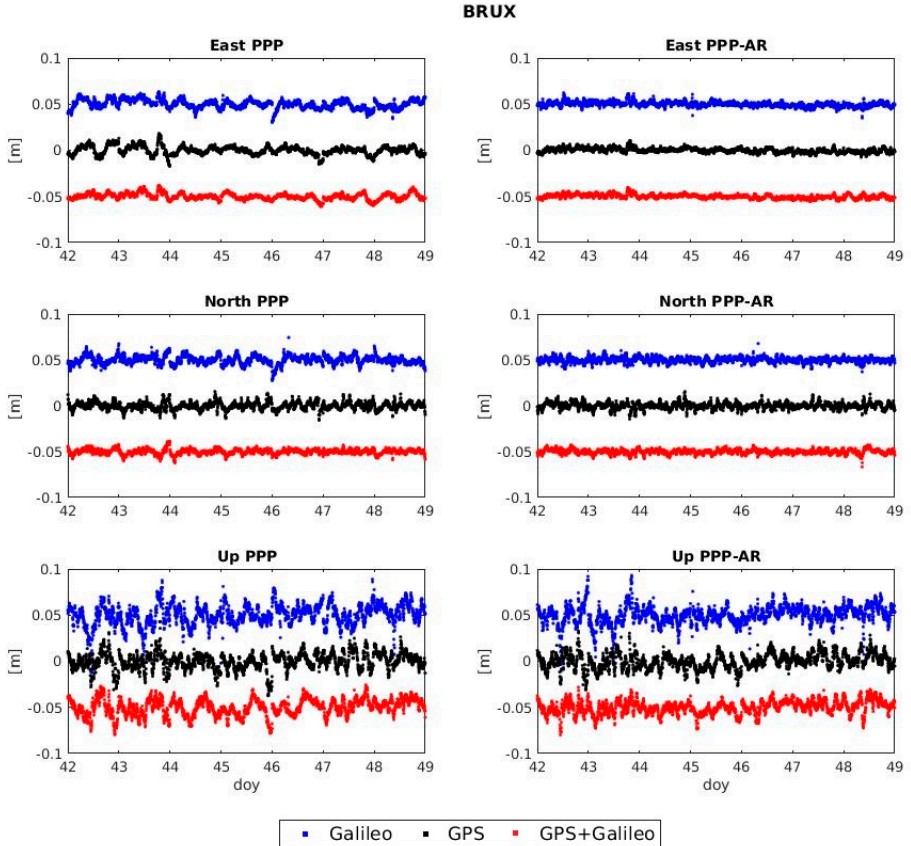

**Figure 2.** PPP and PPP-AR solutions for BRUX station for Galileo-only, GPS-only and Multi-GNSS for the period 11–19 February 2019, DOY: 42–49/2019. For Galileo-only and GPS + Galileo, additional biases of +0.05 m and −0.05 m have been added respectively for better representation in the graph.

**Table 3.** Ambiguity resolution percentages for Galileo and GPS system for BRUX station.

| BRUX AR (%) | 042 | 043 | 044 | 045 | 046 | 047 | 048 |
|---|---|---|---|---|---|---|---|
| Galileo | 95.24 | 100 | 100 | 97.67 | 100 | 100 | 95.45 |
| GPS | 98.63 | 91.67 | 91.30 | 98.44 | 100 | 98.48 | 94.03 |

From Figure 2, it is observed that certain irregularities of the repeatability patterns in the PPP solutions do almost disappear in a PPP-AR solution. There is a little jump at the end of DOY 043 for north and east directions coming from the contribution of the GPS system (also seen in GPS-only PPP of around 2 cm in East and 1 cm in North) to the Multi-GNSS solution. Nevertheless, it is seen that in Multi-GNSS PPP-AR this jump is reduced to less than 0.5 cm.

The importance of AR in precise positioning is also seen when comparing the PPP solution to the PPP-AR solution. It is seen that some jumps in the PPP solution no longer appear when performing PPP-AR processing. The PPP-AR mode is shown to be smoother and more linear than the PPP mode (in particular for the east direction). Another example is the one seen for Galileo-only around DOY 046. There is a downward jump of around 2 cm for east and north directions that is eliminated for the PPP-AR mode.

The 1-σ values of the above temporal series for BRUX and another two examples of stations (CAS1 and NYA) for PPP and PPP-AR are gathered in the following Tables 4–6.

**Table 4.** 1-σ values PPP and PPP-AR for BRUX station.

| BRUX | Mode | East (mm) | North (mm) | Up (mm) |
|---|---|---|---|---|
| Galileo | PPP | 4.7 | 4.6 | 11.7 |
| | PPP-AR | 2.6 | 2.9 | 10.3 |
| GPS | PPP | 4.7 | 4.1 | 9.2 |
| | PPP-AR | 2.4 | 3.4 | 8.5 |
| GPS + Galileo | PPP | 3.4 | 2.7 | 9.1 |
| | PPP-AR | 2.1 | 2.4 | 7.3 |

**Table 5.** 1-σ values PPP and PPP-AR for CAS1 station.

| CAS1 | Mode | East (mm) | North (mm) | Up (mm) |
|---|---|---|---|---|
| Galileo | PPP | 7.1 | 6.8 | 16.6 |
| | PPP-AR | 4.2 | 5.2 | 15.6 |
| GPS | PPP | 6.4 | 6.8 | 15.2 |
| | PPP-AR | 3.8 | 5.2 | 14.3 |
| GPS + Galileo | PPP | 4.6 | 4.5 | 11.3 |
| | PPP-AR | 3.1 | 3.7 | 10.3 |

**Table 6.** 1-σ values PPP and PPP-AR for NYA2 station.

| NYA2 | Mode | East (mm) | North (mm) | Up (mm) |
|---|---|---|---|---|
| Galileo | PPP | 4.9 | 5.0 | 13.9 |
| | PPP-AR | 2.8 | 2.9 | 15.5 |
| GPS | PPP | 4.2 | 4.1 | 16.0 |
| | PPP-AR | 2.4 | 2.2 | 11.7 |
| GPS + Galileo | PPP | 3.3 | 3.1 | 10.3 |
| | PPP-AR | 2.5 | 2.1 | 9.8 |

From these examples it is clear that level of accuracy achieved varies for each individual station. This could be because of several parameters such as multipath, station ANTEX parameters (i.e., for Galileo station, antenna ANTEX files are not yet provided so therefore the GPS station antenna ANTEX files were used), etc. Notwithstanding, it is seen that the level of accuracy from Galileo-only solutions is nearly comparable to the one of GPS-only solutions. For the Galileo-only solutions there were less measurements used than for the GPS-only solution due to the fact that the Galileo constellation has less satellites than GPS. It is expected that once the Galileo constellation is complete, the accuracy of PPP and PPP-AR respectively will improve. Ambiguity resolution improves the solution about 1–2 mm in East and North directions (around 10–45% improvement) (Here and elsewhere in the text, improvements in % are computed according to the formula: $100\% \cdot (v_2 - v_1)/v_1$, where $v_1$ and $v_2$ are the values of before (hence reference) and after the change.) and about 1–2 mm in up direction (around 5–20% improvement). It is observed that the up component is improving less than east and north components when comparing the PPP and the PPP-AR cases using a single GNSS system. This is explained due to the fact that highly correlated parameters: i.e., up component, the tropospheric parameters (i.e., Zenith Tropospheric Delay (ZTD)) and station clocks are better de-correlated in the Multi-GNSS solutions. This happens mainly because more satellite measurements are participating in the LSE solutions.

It is interesting to observe that even though the Galileo-only solution is not better than the GPS-only solution (but it is of the same order), when using both systems the combined solution is improved with respect to the GPS only one. This means that adding Galileo ameliorates the overall performance of positioning both in PPP as well as in PPP-AR mode.

### 3.2. Global Network of Stations

The previous graphs from the example stations give some indications:

- The PPP-AR mode gives better repeatability for the timeseries than the PPP mode.
- Galileo only solution gives similar level of 1-σ values repeatability than the GPS only solution.
- The use of Galileo can improve the current precise positioning situation of GPS when used in a Multi-GNSS combination in a PPP-AR mode.

Nevertheless, the graphs shown in detail referred to a very small specimen of stations. It was considered important to process a network of 50 the IGS stations used in MGEX. In this way it can be investigated whether these assumptions are valid globally and whether there are any potential geographical dependencies that affect the positioning accuracy.

Figures 3 and 4 show the overall network performance of the 1-σ values (as computed for Figure 2) as well and the global RMS values (the RMS of all the values plotted) for GPS PPP-AR and GPS+Galileo PPP-AR scenarios for east, north and up components.

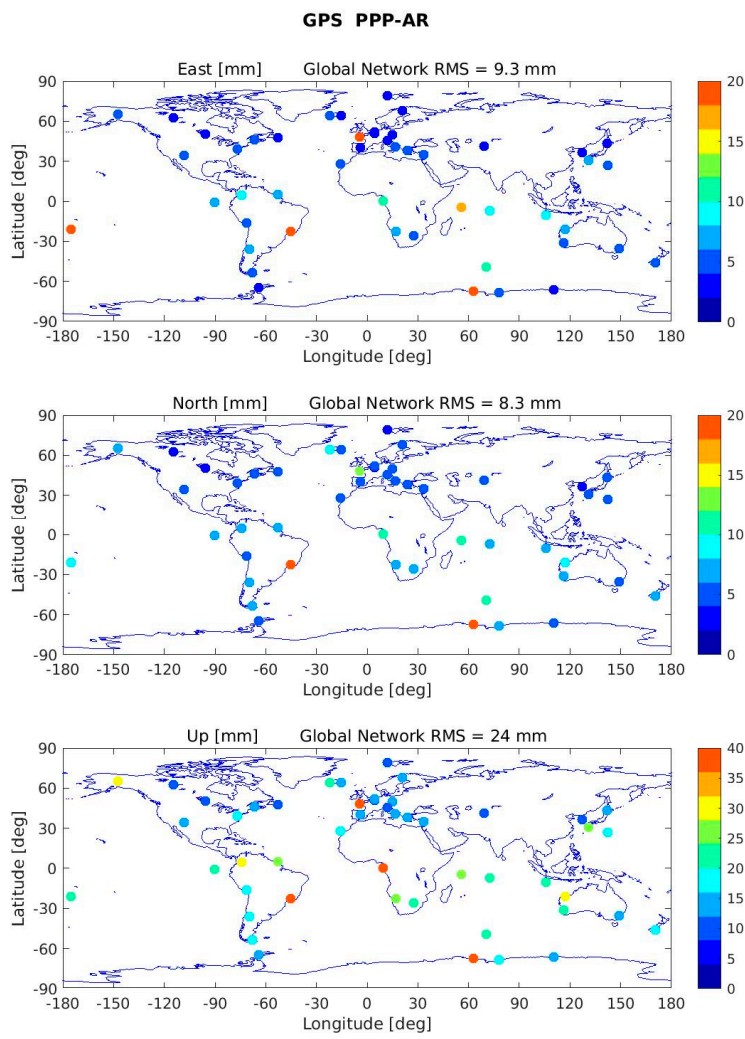

**Figure 3.** GPS PPP-AR solutions for the network of IGS stations in East, North and Up components and their global RMS (mm).

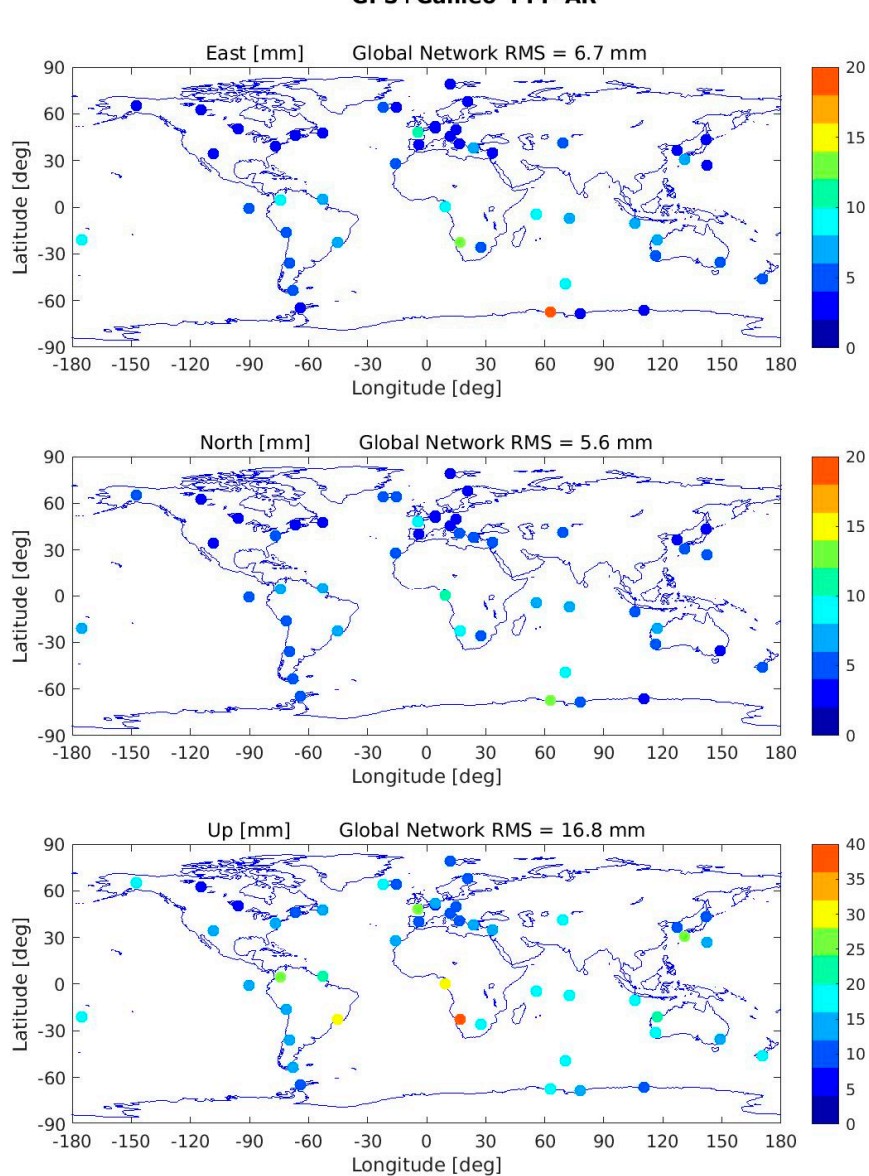

**Figure 4.** GPS + Galileo PPP-AR solutions for the network of IGS stations in East, North and Up components and their global RMS (mm).

The following Table 7 is giving the RMS of all AR percentages for the entire network and the RMS for the whole week of processing. It is observed that in general the Galileo percentages are a little higher than the GPS ones.

**Table 7.** Ambiguity resolution percentages for Galileo and GPS system for BRUX station.

| Network AR (%) | 042 | 043 | 044 | 045 | 046 | 047 | 048 | RMS |
|---|---|---|---|---|---|---|---|---|
| Galileo | 90.71 | 95.38 | 95.39 | 95.56 | 96.16 | 95.67 | 89.39 | 94.07 |
| GPS | 93.55 | 92.25 | 89.85 | 93.90 | 94.09 | 94.54 | 89.62 | 92.56 |

The global positioning RMS values are gathered in Table 8 for the three directions from the global maps. As it is seen, adding Galileo to the constellation can improve the positioning globally both for PPP and PPP-AR modes. Even the PPP mode of Multi-GNSS gives better accuracy than the GPS PPP-AR (which is considered as the best positioning that can be achieved until now). Ambiguity resolution

improves the solution about 1–3 mm in East and North directions (around 10–20% improvement) and about 0–2 mm in Up direction (around 2–8% improvement). Combining Galileo with GPS can improve the timeseries by around 3–4 mm in East and North directions and by 9 mm in Up direction. Comparing the current best positioning performance (i.e., GPS-only PPP-AR) with the one when adding Galileo (i.e., GPS+Galileo PPP-AR), we see that the results improve from 9.3 mm to 6.7 mm (28%) for the East component, 8.3 mm to 5.6 mm (33%) for the North component and 24 mm to 16.8 mm (30%) for the Up component.

**Table 8.** Global RMS of 1-σ values for PPP and PPP-AR for the entire network examined.

| Global | Mode | East (mm) | North (mm) | Up (mm) |
|---|---|---|---|---|
| Galileo | PPP | 17.0 | 14.6 | 33.1 |
| | PPP-AR | 13.7 | 12.2 | 30.8 |
| GPS | PPP | 11.8 | 9.4 | 26.0 |
| | PPP-AR | 9.3 | 8.3 | 24.0 |
| GPS + Galileo | PPP | 7.9 | 6.1 | 17.2 |
| | PPP-AR | 6.7 | 5.6 | 16.8 |

## 4. Discussion and Conclusions

In this paper we investigated the performance (in term of repeatability) of kinematic PPP and PPP-AR positioning using GPS, Galileo and combined measurements of the two systems on a global network.

The Galileo-only PPP-AR solution performance repeatability reaches 14 mm in horizontal direction and 31 mm in vertical direction. This level is of similar order of magnitude and just below the GPS-only solution one. It is expected that once the Galileo constellation is complete, the accuracy of Galileo-only PPP and PPP-AR will improve relative to today (due to the addition of four more satellites and to the availability of receiver antenna calibration for Galileo L5 frequency).

The Multi-GNSS solutions for both PPP and PPP-AR give much better results in term of repeatability than both systems used separately (we observe a gain in repeatability of around 30% in horizontal and in vertical directions); and this even though the Galileo-only solutions are by little not better than the GPS-only solutions. This is logically explained by the increased number of measurements and the highest satellite geometry diversity in the Multi-GNSS solutions relatively to the single systems ones (with only one inter-system bias parameter added). This result proves also that despite the unavailable receiver's antenna patterns for the E5 Galileo frequencies used in this study, the two systems are already compatible at the sub-centimeter level. We assume, as a consequence, that parameters that are highly correlated such as the vertical components, the tropospheric ZTD parameter and the station clocks are more de-correlated in the Multi-GNSS solutions than in single-systems ones.

The best repeatabilities observed in this study for a 300 s dynamic Multi-GNSS GPS + GAL PPP-AR solution reach less than 3 mm in horizontal direction and 7 mm in the vertical one for the BRUX receiver. The global network statistics are a little larger with 8 mm in horizontal and 17 mm in vertical directions. For the global network, no particular geographical pattern has been seen among the 50 stations used.

These results show that Galileo can really contribute to Multi-GNSS precise positioning and improve the best solutions obtained today with GPS. Scientific, geodetic and geophysical applications can already benefit from the combined processing of GPS and Galileo observations.

**Author Contributions:** Conceptualization, G.K., S.L. and F.P.; methodology, G.K.; software, S.L.; G.K. writing—original draft preparation, S.L. and F.P.; writing—review and editing.

**Funding:** This research received no external funding.

**Acknowledgments:** All authors would like to acknowledge the work of Mini Gupta and Jean-Charles Marty for their work changing the EXE-PPP and the GINS software. These were extensively used for the experiments of this article. In addition, fruitful discussions with Flavien Mercier and Alvaro Santamaria were very much appreciated.

**Conflicts of Interest:** The authors declare no conflict of interest.

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
