# Peer review of "PPP and PPP-AR Kinematic Post-Processed Performance of GPS-Only, Galileo-Only and Multi-GNSS"

_remotesensing, doi:10.3390/rs11212477_

Round 1
Reviewer 1 Report
PPP-AR is an important technology, but not a new technology. 1.This study only studied the GPS, Galilieo PPP-AR, but not studiled the GLonass and BDS, this should be also considered. 2. The study only used the post solution, but no real-time solution, this shoule be also considered, as it is more inportant for the PPP application. 3. For PPP-AR, we also care about the convergence time, it should be added.Author Response
We agree that PPP-AR is not a new technology, that GLONASS and Beidou is not considered in this study and that the convergence time issue is not discussed too. However, we explicitly state that “In this paper we examine the results of postprocessed kinematic PPP and PPP-AR using Galileo-only, GPS-only and Multi-GNSS (GPS + Galileo) constellations”. The main reasons why we limitted the discussion to GPS and Galileo only are the following. GPS is being providing the most precise PPP solutions for decades. But since February this year 22 Galileo satellites are usable and the CNES-CLS IGS Analysis Center is providing to the community precise orbits and IRC clocks for those satellites that can thereby be used for PPP-AR. Comparing individual and joint GPS and Galileo PPP-AR performance seemed interesting to us. In fact there is very few literature demonstrating that GPS PPP-AR results can be improved by adding Galileo data.
We also agree that real time PPP has a huge variety of applications. Nevertheless post-processed PPP is more accurate and has many scientific users. A discussion on convergence time is not applicable in this case.

Reviewer 2 Report
In this paper Authors present a topic which is of high interest - the Galileo positioning using PPP and PPP-AR in post-processing mode. Precise products (orbits, clocks) from their own agency are used, and the final conclusions are as expected - the Galileo contributes significantly to the precise positioning. Here where positive aspects end. I regret, but the paper is prepared very disreputably and chaotic, it lacks critical information in methodology and therefore results cannot be interpreted and understood in details by a reader. Such messily prepared papers should be rejected outright, however, due to important contribution this paper can provide, I encourage authors to resubmit the paper after the major and careful revision. Below is the list of (>60) suggested corrections and I am looking forward for detailed answer (and corresponding corrections in the text) for each one.
ABSTRACT
1) Abstract is missing any quantitative indicators on results achieved with Galileo or compared to GPS. Add.
INTRODUCTION
2) Abbreviations in introduction should be explained again, not all readers start with abstract. Explain PPP, AR, CNES-CLS, IGS, ....
3) L33-34: Are code observations also used in zero-difference mode?
4) L36: why static positioning is not consider as scientific application but kinematic is?
5) L40: "calculation" -> determination"
6) L42: is CNES the only AC allowing for PPP-AR? In no, why products have to be taken from CNES? Rephrase or explain. Do you mean "consistent", rather than "coherent"?
7) L43 - there are hundreds of publications showing GPS AR, GLONASS AR, some also with BDS and multi-GNSS. I do not agree with "a few". Maybe real-time is still more challenging, but post-processing PPP-AR is well known for some time already.
8) L44 - PPP has nothing common with RTK, I do not understand neither a meaning nor a logic of this sentence.
9) L46: DD can also be used for daily post-processed positioning. Explain what do you mean here.
10) L50 - please give more details about the current status of Galileo. Are all satellites active and in correct position?
11) Hypothesis - "... will give better accuracy" than? GPS-only? Galileo-only? Be more precise.
12) L53 - not only intersystem bias is added but also Galileo ambiguities. What about multi-frequency case and additional clock biases for GPS L5? Elaborate in the text.
MATERIALS AND METHODS
13) Separate clearly (subsections) the part related with products generation from methods for positioning at the user side.
14) "CNES AC has the advantage" - over what or which ACs? Other products also allow for single-station positioning. Moreover, as you state later, CNES also uses global network for determination of satellite product.
15) L64: give the extension of a file that contains WL satellite biases. Is this a standard format? If not, explain or give reference.
16) L48: "Where" -> "where"
17) in all the following explanations carefully check the use of "is" and "are"; use also semicolon on the end of each line (except the last).
18) 4th dot: move the eq. delta t = dtr-dts to the very beginning
19) 5th and 6th dot - troposphere and ionosphere delay, this is the commonly accepted naming
20) 7th dot - do not use etc., be specific; is relativistic effect a code measurement error or a clock correction?
21) last dot - "code" -> "phase"; do not use etc.; wind up was already mentioned as a separate term - why it is given explicitly? usually all geophysical corrections are included in the geometric term. Please review all the equations very carefully!
22) L96 - "It is useful..." Sentence out of context, remove.
23) Ew. 5 should be introduced just after reference [5] and [6].
24) L107 - remove "In contrast".
25) L110 Do not use phrases like "clearly seen"
26) L113 Explain how u^s is calculated from CNES or give reference
27) L112 Explain what is the benefit (or why it is necessary) to use network processing
28) Nmw,s^s - explain how it is determined.
29) Eq. 6-7 should be introduced after citation [16].
30) L115 "the" -> "then"?
31) L116-117 - please use more straightforward introduction of the iono-free combination.
32) Eq. 8 and 9 are irrelevant, remove.
33) L120 "Where" -> "where"
34) Table 1 - Is this relevant for the paper?
35) L131 - Rephrase the first sentence.
36) L132-133 - How N is estimated as an integer? Details requested.
37) Is W_r^s constant? You wrote that it is "calculated individually once". It is not clear what does it mean.
38) Figure 1 - in fact there are two schemes (or a link is missing). Please split into two figures or use a) and b).
39) Explain "bootstrapping", this is not a common term
40) Table 2:
a) is this for product determination or user-side processing? Or both?
b) missing units for weighting; also explain why such numbers or give reference - they are not very typical
c) which elevation weighting function was used?
d) you give reference to GPT2 but not for VMF, why? which exactly VMF did you use? 1 or 3? final or predicted? why not merging GPT2 with GPT or use everything from VMF? your approach is quite strange
e) Kouba [26] for GPS and GSA [22] for Galileo
f) reference to FES 2012 should be line above than it is now
g) missing reference to IERS 2010
h) explain C04
i) what is [26], at least a word is requested here
j) gradients (E,N) are not present in Eq. 1-4; which mapping function was used for gradients (I know at least 3)? did you estimate total or wet delay only? if wet, what was the hydrostatic model?
k) did you estimate N,E,U directly(how)?! or you did transformation from XYZ to topocentric?
41) L148 - "These corrections" - remove, this is obvious
42) L149 - elaborate which parameters are provided in igs14.atx for Galileo
43) Missing in this section: station map (location)
44) Missing in this section: information about modulation (L1C, L1W, L1X, ...). As far as I know, the consistent use of measurement and products is relevant for ambiguity resolution. If not, explain.
RESULTS
45) General comment - too many similar figures and tables, without a clear idea how to present results.
a) For figures 2-6 you can avoid it e.g by using colors N,E,U and present 3 time-series on one plot and 3 histograms on the right panel. Instead of 36 you will have 12 (which is still a lot). Consider combination of N,E to Horizontal component.
b) Please consider how to combine figure 8-10 into one figure (maybe use 3 columns?) Explain doy (DoY), give year. E,N and U abbreviations were not explained before
c) There are two Tables 4
d) Table 4,4 and 5 should be combined into one table. Why these particular stations were selected?
e) Figure 11-16 are "hard to read". Because you did not found any geographical pattern, maybe it's better to replace them with bars? Consider showing horizontal component instead of N,E to reduce the length of the paper. Now it looks more like a report (give as much as you can, the more the better), but this is not really expected from the well written paper.
46) It is not explained what is the reference (truth).
Please note that I do not comment on results achieved, because of the missing critical information in methodology description and my request for different presentation of results. I will comment in the next review round, if the requested improvements are introduced. Moreover, I realized you never said about the reference coordinates (IGS official solution)? 1-cm level accuracy in kinematic PPP is extremely optimistic. I thought that maybe these are formal errors (from Cx matrix), but your values are also negative. What happened with the initialization time - did you use backward smoothing? I hope you understand, that there are so many unknowns in the paper, that it cannot be evaluated correctly.
Finally, the description of the results is very immature, even amateur. Please rewrite this section.
DISCUSSION AND CONCLUSIONS
47) Multi-GNSS is usually used for >2 systems
48) "This can be explained..." (..) "... degree of freedom" - this is too long and very "infantile".
49) L302: name tropospheric parameters which are correlated with Up and receiver clock
50) L307 - better means more accurate, precise or both?
REFERENCES
Style should be consistent, please carefully check journal requirements.
51) Many missing journal numbers (eg. 3, 4, 5) and p ages (eg. 15, 24)
52) What is [20], is this peer reviewed?
53) [19] is m issing the access date (I am not sure about the journal style, but usually an exact date should be given, the same for [17])
54) Please use consequently "" or << >>
55) In 28 there are some strange symbols after Mervart
56) [17] do not use capitals
57) There are too many proceedings (usually not peer reviewed). From good quality paper it is expected that you cite more journal publications. Please change when possible.
58) Regarding the performance of Galileo (as stated in the title) recently a paper was published in GPS Solutions. Please check their results, consider mentioning in the introduction or discussion if possible.
59) You also cite quite a lot of French papers in the introduction (11, when 28 is total). Can you also give credits to international authors? Especially when you give examples of PPP application more than one author can be used.
60) [6] isn't Krzysztof a name? Please double check, this is your own (!) paper...
61) Missing references (and some comments in the text) about other results achieved for PPP (and PPP-AR if such exist) in the kinematic mode, not only for Galileo but also for GPS. Are your results comparable or superior to results presented in other studies? Are they comparable (the same measures to assess the performance)? See also my comment 58) in this context.
Author Response
ABSTRACT
1) Abstract is missing any quantitative indicators on results achieved with Galileo or compared to GPS. Add. -> Abstract changed to give more quantitative indicators.
INTRODUCTION
2) Abbreviations in introduction should be explained again, not all readers start with abstract. Explain PPP, AR, CNES-CLS, IGS, .... à corrected
3) L33-34: Are code observations also used in zero-difference mode? ->Well they are used for the PPP, but for the PPP_AR we use only the Ionosphere free carrier phase fixed measurements. We rephrased to “zero-difference observations” to avoid any confusion. Further details are given about the processings of PPP and PPP-AR above Figure 1.
4) L36: why static positioning is not consider as scientific application but kinematic is? The sentence is rephrased to: “It has been used in numerous scientific applications; namely general static positioning, kinematic positioning, time transfer etc.”
5) L40: "calculation" -> determination" à corrected
6) L42: is CNES the only AC allowing for PPP-AR? In no, why products have to be taken from CNES? Rephrase or explain. Do you mean "consistent", rather than "coherent"? At the time of writing this article the CNES/CLS AC is the only IGS AC that provides IRC products and satellite biases for the Galileo system that can allow for Galileo-only PPP-AR. Satellite biases and IRC products are needed to perform PPP-AR. The sentence rephrased to be more clear: “The PPP with ambiguity resolution processing has the advantage that the processing can be done directly using only one station; i.e. there is no need for a global network of stations. The requirement, however, in order to be able to do such processing is to use the consistent products. In this study products are taken from the CNES-CLS AC:’
7) L43 - there are hundreds of publications showing GPS AR, GLONASS AR, some also with BDS and multi-GNSS. I do not agree with "a few". Maybe real-time is still more challenging, but post-processing PPP-AR is well known for some time already. à corrected
8) L44 - PPP has nothing common with RTK, I do not understand neither a meaning nor a logic of this sentence.-> you are right. The Technique of RTK is different than PPP even though the AR is known to be useful to RTK. This sentence got removed.
9) L46: DD can also be used for daily post-processed positioning. Explain what do you mean here.-> sentence rephrased : “In addition, PPP and PPP-AR are some of the methods that can be used for daily post-processed positioning” We didn’t want to impy that this was the only way, but some of the ways.
10) L50 - please give more details about the current status of Galileo. Are all satellites active and in correct position? à corrected. Phrase added : « As of 2019, the Galileo constellation comprises of 26 satellites in total (4 In-Orbit Validation (IOV) and 22 Full Operational Capability (FOC)). Two of the FOC satellites are in eccentric orbits, one IOV is not available and one FOC is not usable. The total number of usable satellites is 22: 3 IOV and 21 FOC satellites [9]. »
11) Hypothesis - "... will give better accuracy" than? GPS-only? Galileo-only? Be more precise. ->Sentence changed to: “The following hypothesis is examined: if GPS and Galileo measurements are of the same quality, then the simultaneous use of the two systems in PPP-AR will give better station repeatability than when using each system separately.”
12) L53 - not only intersystem bias is added but also Galileo ambiguities. What about multi-frequency case and additional clock biases for GPS L5? Elaborate in the text. -> We do not compute GPS L5. We are combining GPS L1/L2 with Galileo E1/E5a. This information appears later in the details of the method. Furthermore, in the case of PPP-AR the Galileo ambiguities are resolved so there are no longer unknown parameters.
MATERIALS AND METHODS
13) Separate clearly (subsections) the part related with products generation from methods for positioning at the user side.-> This article refers to the users side… for the calculation of satellite orbits and IRC clock products we cite another article that explains the methodology of POD (product generation). Description of the processing and the Fig. 1 has been corrected and more detailed (answer also to 40)a)
14) "CNES AC has the advantage" - over what or which ACs? Other products also allow for single-station positioning. Moreover, as you state later, CNES also uses global network for determination of satellite product. à Yes other AC can allow for single-station. But for Galileo-only PPP-AR, the CNES/CLS AC is (so far) the only one, because it provides IRC clocks and WL satellite biases. As far as we know there is no other AC that provides such information on a regular basis on IGS server products area. Regarding the network this phrase got removed and added to the global network os stations examination. It was misleading in the method section. Basically the single station can be done and we tested several stations.
15) L64: give the extension of a file that contains WL satellite biases. Is this a standard format? If not, explain or give reference. àPhrase written : …and the Wide-Lane satellite biases (.wsb) “(given either in the header of the clock files or taken from the CNES/CLS portal: ftp://ftpsedr.cls.fr/pub/igsac/ Wide_lane_GAL_satellite_biais.wsb or Wide_lane_GPS_satellite_biais.wsb) The format is self explenatory in the header of the files.
16) L48: "Where" -> "where"à corrected
17) in all the following explanations carefully check the use of "is" and "are"; use also semicolon on the end of each line (except the last). à corrected, the semicolon might not be necessary, please refer to the manuscript template about bulleted lists
18) 4th dot: move the eq. delta t = dtr-dts to the very beginningà corrected
19) 5th and 6th dot - troposphere and ionosphere delay, this is the commonly accepted namingà corrected
20) 7th dot - do not use etc., be specific; is relativistic effect a code measurement error or a clock correction? àYes you are right, this was an error: text has been corrected for code and for phase measurements errors parts.
21) last dot - "code" -> "phase"; do not use etc.; wind up was already mentioned as a separate term - why it is given explicitly? usually all geophysical corrections are included in the geometric term. Please review all the equations very carefully! à corrected. In our equations we have it as a separate term.
22) L96 - "It is useful..." Sentence out of context, remove. à corrected
23) Ew. 5 should be introduced just after reference [5] and [6]. à corrected
24) L107 - remove "In contrast".à corrected
25) L110 Do not use phrases like "clearly seen"à corrected
26) L113 Explain how u^s is calculated from CNES or give referenceà references given
27) L112 Explain what is the benefit (or why it is necessary) to use network processingà sorry this phrase was a mistake. It is removed
28) Nmw,s^s - explain how it is determined.-> Phrase added “In order to be able to separate the and the from the a system of equations (like Eq. 5) is formed and solved by using a Least Squares Estimation (LSE) processing with a bootstrap method [17] [18]. The result after the LSE solution are per satellite pass and ”
29) Eq. 6-7 should be introduced after citation [16]. à corrected
30) L115 "the" -> "then"? -> sentence removed
31) L116-117 - please use more straightforward introduction of the iono-free combination. -> Phrase added: ionosphere-free linear combination that has the property to calcel out the first order of ionospheric effects.
32) Eq. 8 and 9 are irrelevant, remove. à corrected
33) L120 "Where" -> "where"à corrected
34) Table 1 - Is this relevant for the paper? àYes, it explains the values of wide-lane and narrow-lane wavelengths for the specific frequencies for GPS and Galileo used in this paper. We wish to keep it.
35) L131 - Rephrase the first sentence. The term is calculated by using the models proposed by Kouba
36) L132-133 - How N is estimated as an integer? Details requested.-> Phrase added “All equations are then gathered to form another LSE processing. The system of equations for the ionosphere_free code and phase measurements are modelled according to the models given in Table 2 and used to estimate all the parameters: stations position, tropospheric parameters, receiver clocks and inter system bias in case of multi GNSS measurements. For the PPP case the Ν_(r,i)^s are solved with the others parameters as real values. For the PPP-AR case they are fixed to integer numbers within a bootstrap processing
37) Is W_r^s constant? You wrote that it is "calculated individually once". It is not clear what does it mean.
Rephrased The term is calculated by using the models proposed by Kouba
38) Figure 1 - in fact there are two schemes (or a link is missing). Please split into two figures or use a) and b). -> No. Actually they are one figure. They explain the steps needed for the PPP and the PPP-AR processing. For example, in order to do PPP-AR it is needed to complete the pre-processing and the PPP steps. More explanations are given in the text above the figure.
39) Explain "bootstrapping", this is not a common term. -> The (one per epoch of measurements) and the (one per satellite pass) are solved using all available equations (Eq. 5 corrected by the ) with a Least Squares Estimation (LSE) processing associated with a bootstrap method [17] [18]. The result after this step are the and
40) Table 2:
a) is this for product determination or user-side processing? Or both?-> it is user side processing b) missing units for weighting; also explain why such numbers or give reference - they are not very typical->units given. These numbers are studied and used from the CNES/CLS AC since a long time, and there is no reference. Basically it is a way to give more importance to the phase measurements and less to code measurements. This is the reason behind such numbers.
c) which elevation weighting function was used? ->Extra row in the table to give the elevation function used.
|
Elevation weighting function |
d) you give reference to GPT2 but not for VMF, why? which exactly VMF did you use? 1 or 3? final or predicted? why not merging GPT2 with GPT or use everything from VMF? your approach is quite strange ->
A priori local meteorological parameters (pressure, temperature, and wet mapping function coefficients) of GPT2 model are used to compute hydrostatic delays and for the wet mapping function VMF1.We then adjust 1 zenithal tropospheric delay per two hours in factor of the wet mapping function.
e) Kouba [26] for GPS and GSA [22] for Galileo à corrected f) reference to FES 2012 should be line above than it is nowà corrected, line about tide models is irrelevant and got removed g) missing reference to IERS 2010àadded h) explain C04àcitation added i) what is [26], at least a word is requested hereà corrected j) gradients (E,N) are not present in Eq. 1-4; which mapping function was used for gradients (I know at least 3)? did you estimate total or wet delay only? if wet, what was the hydrostatic model? (1 couple of gradients in north and East direction are also adjusted per day following ( IERSConventions (2010) section 9.2, Chen & Herring 1997) added in Table 2. k) did you estimate N,E,U directly(how)?! or you did transformation from XYZ to topocentric?-> No we estimated X,Y,Z and then we transferred them to E,N,U, phrase added to the table. “X, Y, Z transformed to East, North, Up”41) L148 - "These corrections" - remove, this is obviousà corrected
42) L149 - elaborate which parameters are provided in igs14.atx for Galileoà The IGS is compiling a consistent set of absolute Antenna Phase Centre (APC) (i.e. Phase Center Offset (PCO) and Phase Center Variation(PCV))
43) Missing in this section: station map (location) à in this section there is no need for a map. PPP and PPP-AR are done for single stations. Later when we perform PPP and PPP-AR for 50 IGS MGEX stations we show two examples for E,N, and U. But at this point we do not consider a map would be necessary.
44) Missing in this section: information about modulation (L1C, L1W, L1X, ...). As far as I know, the consistent use of measurement and products is relevant for ambiguity resolution. If not, explain. ->We do not use any additional bias for Galileo ambiguity fixing since a previous study (G. Katsigianni & al., "Improving Galileo orbit determination using zero-difference ambiguity fixing in a Multi-GNSS processing," Advances in Space Research, no. 63, p. 2952–2963, 2019.) showed that WSB Galileo biases were compatible will all kind of receiver and modulations. Moreover, this paper proves that it is not necessary because we are fixing ambiguities on all stations tested.
RESULTS
45) General comment - too many similar figures and tables, without a clear idea how to present results. -> corrected
a) For figures 2-6 you can avoid it e.g by using colors N,E,U and present 3 time-series on one plot and 3 histograms on the right panel. Instead of 36 you will have 12 (which is still a lot). Consider combination of N,E to Horizontal component. ->For the combination of these graphs we followed the recommendation of reviewer#3. Please refer to his comment b) Please consider how to combine figure 8-10 into one figure (maybe use 3 columns?) Explain doy (DoY), give year. E,N and U abbreviations were not explained beforeà These figures are removed as they are the same as the previous graphs…they are ‘condensed’ with the previous graphs in one plot. Please refer to the recommendations of reviewer #3 c) There are two Tables 4à corrected d) Table 4,4 and 5 should be combined into one table. Why these particular stations were selected? à We consider it is better to leave the Tables as they are. Bar plots might be confusing. The selection of these station was hazardous. Just to show two more examples. Later a bigger network is presented for 50 stations to show that the method is generally working well regardless the station. e) Figure 11-16 are "hard to read". Because you did not found any geographical pattern, maybe it's better to replace them with bars? Consider showing horizontal component instead of N,E to reduce the length of the paper. Now it looks more like a report (give as much as you can, the more the better), but this is not really expected from the well written paper. ->We did try to make a bar graph but it was complicated. We therefore decided to remove the figures but keep only two: the GPS PPP-AR and the GPS+Galileo PPP-AR. Like this there are less figures for the readers and the point that the method is generally working well for all the stations examined. For the other cases the statistics in the Table for the Global RMS are kept for a means of comparison.46) It is not explained what is the reference (truth). -> We add in the text:
For each temporal serie the mean value is calculated and subtracted to center the serie to zero before computing the statistics (and for all statistics computed in this study). This mean that the reference is not the same for each serie. We focus here only on the repeatability of the estimated coordinates. The constant biaises between the solutions (generally not significant) are ignored in this study. For Galileo only and GPS+Galileo additional biases of +0.05 and -0.05 have been added respectively for better representation in the graph.
Moreover, I realized you never said about the reference coordinates (IGS official solution)? 1-cm level accuracy (repeatability!) in kinematic PPP is extremely optimistic. I thought that maybe these are formal errors (from Cx matrix), but your values are also negative. -> please refer to the comment above (46). For each timeserie the mean value is removed. We show repeatability numbers and not accuracy numbers. Furthermore, the IGS official solution is using GPS system only. And in addition for some stations the ANTEX files are still missing, so there could be some biases compared to the IGS official solution.
What happened with the initialization time - did you use backward smoothing? -> no we did not
Finally, the description of the results is very immature, even amateur. Please rewrite this section. Conclusion was completely revisited
DISCUSSION AND CONCLUSIONS
47) Multi-GNSS is usually used for >2 systems -> We beg to differ : multi- is a prefix for ‘multiple’ which is the opposite of ‘single’….so more than one is considered as multi- Reviewer #3 suggests that we include a term “multi-GNSS” in the title too.
48) "This can be explained..." (..) "... degree of freedom" - this is too long and very "infantile". -> Rephrased to “This can be explained due to the fact that with Multi-GNSS solution there is nearly double the number of measurements of similar good positioning quality. For the system of equations this means that there are nearly double the number of measurements with adding only one extra unknown parameter; the inter-system bias. Overall the system of equations has a lower degree of freedom. As a consequence, parameters that are highly correlated such as the Up component, Tropospheric parameters (i.e. Zenith Tropospheric Delay (ZTD)), Integer ambiguities, station and satellite clocks can be more easily de-correlated and thus, more accurately estimated.”
49) L302: name tropospheric parameters which are correlated with Up and receiver clock-> added, ZTD
50) L307 - better means more accurate, precise or both? -> Better repeatability, corrected.
REFERENCES
Style should be consistent, please carefully check journal requirements.
51) Many missing journal numbers (eg. 3, 4, 5) -> what do you mean here? Issue numbers? If yes, please refer to the template, we do not think that they are required. However we do give the doi numbers whenever it is possible for more clarity, and pages (eg. 15, 24) -> Katsigianni et al. is quite new and does not yet have page numbers, for Hernandez-Pajares et al. we did include the page number
52) What is [20], is this peer reviewed? à This is the manual of GINS software. It is not peer reviewed.
53) [19] is missing the access date (I am not sure about the journal style, but usually an exact date should be given, the same for [17]) -> corrected
54) Please use consequently "" or << >>-> corrected
55) In 28 there are some strange symbols after Mervart-> corrected
56) [17] do not use capitalsà corrected
57) There are too many proceedings (usually not peer reviewed). From good quality paper it is expected that you cite more journal publications. Please change when possible. à there are 5 conference proceedings, but unfortunately there are no corresponding journal articles. Sorry
58) Regarding the performance of Galileo (as stated in the title) recently a paper was published in GPS Solutions. Please check their results, consider mentioning in the introduction or discussion if possible. à We were not aware of this paper at the time of writing this article. This reference has been added in the introduction.
59) You also cite quite a lot of French papers in the introduction (11, when 28 is total). Can you also give credits to international authors? Especially when you give examples of PPP application more than one author can be used. à we added some more international references in the introduction section.
60) [6] isn't Krzysztof a name? à corrected
61) Missing references (and some comments in the text) about other results achieved for PPP (and PPP-AR if such exist) in the kinematic mode, not only for Galileo but also for GPS. Are your results comparable or superior to results presented in other studies? Are they comparable (the same measures to assess the performance)? See also my comment 58) in this context. à we added some other references but we couldnt find other articles that deal with GPS + Galileo PPP-AR time series repeatability results.
Reviewer 3 Report
Review report of “PPP and PPP-AR kinematic post-processed performance of GPS and Galileo” by Katsigianni et al.
I accepted the responsibility of reviewing this paper by Katsigianni et al. because the paper aimed to evaluate the kinematic post-processing performances for GPS, Galileo and GPS+Galileo constellations using PPP and PPP-AR method. As I started reading this paper with great interest, I realized that, in some places, additional details were required. In the introduction section, authors should present some background with reference to discuss how PPP-AR was used and validated through numerous applications as claimed in line 17 -18 in the abstract. The authors should also provide some insights on Katsigianni et al., (2019) that used a similar method and compared the GPS-only and Galileo-Only solution. The authors also discuss how this study compares to the previous one and what improvement in made in this study.
Also, readers may find the result section confusing, given that there is repeated information presented on multiple graphs (Figure 2 -7 vs Figure 8 - 10). Authors need to find a way to summarize these graphs to avoid consuming space. Same comments for the maps.
The Discussion and Conclusion section must to rewritten so that it answers all the research questions. A detail discussion is necessary to support the findings mentioned in the result section.
I would recommend that all authors revise this manuscript carefully and provide details where requested before resubmitting.
There are some specific comments that may help the authors towards improving the manuscript.
Specific Comments
Line 2 – 3: The manuscript described the multi-GNSS solution as well. Thus, I recommend rewriting the title to add multi-GNSS as mentioned in line 18 -19 in the abstract.
Introduction:
Line 43 – 44: The authors should also present a comparative discussion focusing on the novelty of the method using in this study.
Result:
The result section must be improved.
Figure 2 – 10: These figures need to be summarized. Figure 8 -10 seems to be repetitions of figure 2 -7. Two suggestions are given below which could replace fig 2 - 10.
Authors could make a plot with 6 subplots (3 x 2) where the left column could show PPP solutions and the right column could show PPP-AR solution. Each row could show E, N, and U. Each subplot could show solutions for GPS, Galileo, and GPS+Galileo. For example, in 1 x1 subplot, authors could plot PPP solutions for GPS, Galileo, and GPS+Galileo where two of the three solutions can be plotted with some arbitrary offset to avoid the overlap. Another option would be making 3 figures instead of one (making separated figure for E, N and U component). Since this study is comparing PPP and PPP-AR solutions for three different solutions it would be helpful to have them together, given that the study does not compare results among E, N and U component. Thus, the author could make a 3 x 2 plots where columns are for PPP and PPP-AR and rows are for GPS, Galileo, and GPS+Galileo solution for one component. Same can be done for the other two components.
Table 4 – 6: Table 5 is mislabeled as table 4. Readers might find it difficult to compare the result presented in table 4 – 6. Thus, I would recommend that authors present this result in a grouped bar graph.
Line 228 – 230: Author should explain why up (U) components improve more than the other two.
Line 231 – 234 and Table 4 -6 (Up component column): Author should provide a possible explanation in the discussion section why Galileo-only solution is not performing better than GPS-only solutions and why multi-GNSS solutions perform better. Authors should also discuss how these results compare to the results presented in Katsigianni et al., (2019) (Galileo millimeter-level kinematic precise point positioning with ambiguity resolution) that claimed to observed higher precision on Galileo-only solution compare to GPS-only solution (4thparagraph of summary and conclusion of Katsigianni et al., (2019)).
Figure 11 – 16: The readers may find these figures confusing. I would recommend moving these figures into supplementary materials and provide a bar graph instead which could also replace table 7.
Discussion and Conclusion:
This section needs to be rewritten carefully and in an organized manner so that the readers find the answers (in the order) that were promised to deliver in the abstract and introduction section.
Line 294 – 296: The authors are claiming that the quality of positioning is similar for the two systems although earlier in the manuscript (Line 231 – 234 and the tables with results), authors claimed that Galileo-only solution was no better than GPS-only solution. Authors should rewrite this sentence for clarity and avoid contradiction.
Line 305 – 306: This statement is vague. Are authors mentioning that it will improve compare to present Galileo only solutions?
Line 307 – 309: same as line 231 – 234. Delete or rewrite.
Line 310: Readers may find this statement confusing as this line contradict with the results presented in table 7 and even in table 4 - 6. The RMS improved more East and North component.
Line 311: Rewrite this sentence for clarity. Clarify that Galileo along does not improve but improves when added with GPS.
Reference:
Katsigianni, F. Perosanz, S. Loyer and M. Gupta, "Galileo millimeter-level kinematic precise point positioning with ambiguity resolution," Earth, Planets and Space, no. 71, 12 July 2019
Author Response
Specific Comments
Line 2 – 3: The manuscript described the multi-GNSS solution as well. Thus, I recommend rewriting the title to add multi-GNSS as mentioned in line 18 -19 in the abstract. -> title changed a little “PPP and PPP-AR kinematic post-processed performance of GPS-only, and Galileo-only and Multi-GNSS”
Introduction:
Line 43 – 44: The authors should also present a comparative discussion focusing on the novelty of the method using in this study. -> the novelty is presented in the previous article we published in 2019 about Galileo POD. In this article we describe the performance of PPP and PPP-AR when using the satellite and clock products. It is one step further. Citations of our previous works are given in the introduction section. “In a previous study [6] we presented the zero-differences approach used to compute IRC products for Galileo and GPS. Recently we presented the positioning capabilities of Galileo-only solutions [10]. In this article we examine the performance and accuracy of PPP and PPP-AR in a comparison of Galileo-only, GPS-only and combined PPP and PPP-AR solutions.”
Result:
The result section must be improved.
Figure 2 – 10: These figures need to be summarized. Figure 8 -10 seems to be repetitions of figure 2 -7. Two suggestions are given below which could replace fig 2 - 10. à Figures 2-7 are grouped into one and Figure 8-10 are removed.
Authors could make a plot with 6 subplots (3 x 2) where the left column could show PPP solutions and the right column could show PPP-AR solution. Each row could show E, N, and U. Each subplot could show solutions for GPS, Galileo, and GPS+Galileo. For example, in 1 x1 subplot, authors could plot PPP solutions for GPS, Galileo, and GPS+Galileo where two of the three solutions can be plotted with some arbitrary offset to avoid the overlap. -> we followed your recommendation. Thank you !
Another option would be making 3 figures instead of one (making separated figure for E, N and U component). Since this study is comparing PPP and PPP-AR solutions for three different solutions it would be helpful to have them together, given that the study does not compare results among E, N and U component. Thus, the author could make a 3 x 2 plots where columns are for PPP and PPP-AR and rows are for GPS, Galileo, and GPS+Galileo solution for one component. Same can be done for the other two components. -> we followed your previous recommendation.
Table 4 – 6: Table 5 is mislabeled as table 4. Readers might find it difficult to compare the result presented in table 4 – 6. Thus, I would recommend that authors present this result in a grouped bar graph. à We consider it is better to leave the Tables as they are. Bar plots might be confusing. The selection of these station was hazardous. Just to show two more examples. Later a bigger network is presented for 50 stations to show that the method is generally working well regardless the stations processed.
Line 228 – 230: Author should explain why up (U) components improve more (less!!!) than the other two. -> Phrase added there: “It is observed that the Up component is improving more than East and North components. This is explained due to the fact that parameters that are highly correlated: i.e. Up component, the tropospheric parameters (i.e. Zenith Tropospheric Delay (ZTD)) and station clocks are better de-correlated in the Multi-GNSS solutions. This happens because more measurements are used in the Multi-GNSS solutions.” Phrase added in the discussion and conclusions section: “We assume, as a consequence, that parameters that are highly correlated such as the vertical components, the tropospheric parameters (i.e. Zenith Tropospheric Delay (ZTD)),) and the station clocks are more de-correlated in the Multi-GNSS solutions than in single-systems ones.”
Line 231 – 234 and Table 4 -6 (Up component column):
Author should provide a possible explanation in the discussion section why Galileo-only solution is not performing better than GPS-only solutions and why multi-GNSS solutions perform better. -> Phrase added: “For the Galileo-only solutions there were less measurements used than for the GPS-only solution due to the fact that the Galileo constellation has less satellites than GPS.”
Authors should also discuss how these results compare to the results presented in Katsigianni et al., (2019) (Galileo millimeter-level kinematic precise point positioning with ambiguity resolution) that claimed to observed higher precision on Galileo-only solution compare to GPS-only solution (4thparagraph of summary and conclusion of Katsigianni et al., (2019)). -> Maybe you are mistaken, this publication does not compare the Galileo-only to the GPS-only solution. It is a brief letter that shows the positioning capabilities of Galileo-only solutions for PPP and PPP-AR mode in 30 sec post-processed kinematic positioning. The goal of that publication was to show what could be the Galileo-alone positioning repeatability performances. The phrase in the 4th paragraph: “This is a first indication showing that Galileo-only solutions can reach unprecedented levels of precision that can be used for the most high-accuracy demanding post-processing applications.” Is just an indication of the future possibilities of Galileo-alone. We did not imply anywhere that Galileo gives higher precision than GPS-alone. Besides, Galileo as a constellation is not yet completed…
In the present article however we wanted to examine the comparison between GPS-only, Galileo-only and their combinations mainly to show what could be the improvement of the current best positioning (i.e. until now was GPS PPP-AR) with the future one when adding Galileo (i.e. Multi-GNSS PPP-AR)
Figure 11 – 16: The readers may find these figures confusing. I would recommend moving these figures into supplementary materials and provide a bar graph instead which could also replace table 7. ->We did try to make a bar graph but it was complicated. We therefore decided to remove the figures but keep only two: the GPS PPP-AR and the GPS+Galileo PPP-AR. Like this there are less figures for the readers and the point that the method is generally working well for all the stations examined. For the other cases the statistics in the Table for the Global RMS are kept for a means of comparison.
Discussion and Conclusion:
This section needs to be rewritten carefully and in an organized manner so that the readers find the answers (in the order) that were promised to deliver in the abstract and introduction section. - > We tried to rewrite this section again
Line 294 – 296: The authors are claiming that the quality of positioning is similar for the two systems although earlier in the manuscript (Line 231 – 234 and the tables with results), authors claimed that Galileo-only solution was no better than GPS-only solution. Authors should rewrite this sentence for clarity and avoid contradiction. -> Rephrased to: “The Galileo only PPP-AR solution performance repeatability reaches 14 mm in horizontal direction and 31 mm in vertical direction. This level is of similar order of magnitude and just below the GPS-only solution one.”
Line 305 – 306: This statement is vague. Are authors mentioning that it will improve compare to present Galileo only solutions?-> Rephrased to: It is expected that once the Galileo constellation will be complete, the accuracy of Galileo-only PPP and PPP-AR will improve relatively to today (due to the addition of 4 more satellites and to the improvement of receiver antenna calibration for Galileo L5 frequency).
Line 307 – 309: same as line 231 – 234. Delete or rewrite. -> phrase deleted.
Line 310: Readers may find this statement confusing as this line contradict with the results presented in table 7 and even in table 4 - 6. The RMS improved more East and North component.-> phrase changed: “ The Multi-GNSS solutions for both PPP and PPP-AR give much better results in term of repeatability than both systems used separately (we observe gain in repeatability of XXX% in horizontal and YYY% in vertical directions);”
Line 311: Rewrite this sentence for clarity. Clarify that Galileo along does not improve but improves when added with GPS.-> Rephrased: “These results are showing that Galileo can really contribute to Multi-GNSS precise positioning and improve best solutions obtained today with GPS.”
Round 2
Reviewer 1 Report
it is okAuthor Response
Reviewer #1
it is ok -> Thank you
Reviewer 2 Report
The paper has been significantly improved. I am satissfied with most of the answers. Some of my requests were in contrast with Rev #3 (so authors could decide which way do they follow) or are no longer valid due to other major changes. Now the paper present the expected quality (in fact many of previous serious but basic drawbacks should be corrected by authors even before the first submission). I suggest only some minor comments:
Line 80 In the pdf the portal link looks like ftpsedr.cls.fr/pub/igsac/ Wide_lane_GAL_satellite_biais.wsb and it is impossible to reach it. Please double check the link when generating a pdf (I guess: the space before W is the problem and ftp:// is missing). How about "files ...GAL...wsb and ...GPS...wsb available at ftp://ftpsedr.cls.fr/pub/igsac/"
Line 154 ":" should be replaced by ".", otherwise do not start with capital letter
Line 197 "From the above figure" -> " From figure 2"
Line 247 "The following figures (See Figure 3 to Figure 4)" -> "Figure 3 and Figure 4"
Answer to comment 44: I accept your answer, but please give also a short explanation in the text (maybe it is already there, but I couldn't find it).
Author Response
Reviewer #2
The paper has been significantly improved. I am satisfied with most of the answers. Some of my requests were in contrast with Rev #3 (so authors could decide which way do they follow) or are no longer valid due to other major changes. Now the paper presents the expected quality (in fact many of previous serious but basic drawbacks should be corrected by authors even before the first submission). I suggest only some minor comments:
Line 80 In the pdf the portal link looks like ftpsedr.cls.fr/pub/igsac/ Wide_lane_GAL_satellite_biais.wsb and it is impossible to reach it. Please double check the link when generating a pdf (I guess: the space before W is the problem and ftp:// is missing). How about "files ...GAL...wsb and ...GPS...wsb available at ftp://ftpsedr.cls.fr/pub/igsac/" -> Thank you we corrected the phrase to : “(given either in the header of the clock files or taken from the CNES/CLS portal. Files Wide_lane_GAL_satellite_biais.wsb and Wide_lane_GPS_satellite_biais.wsb are available at ftp://ftpsedr.cls.fr/pub/igsac/)”, and now the link is working also at the pdf version.
Line 154 ":" should be replaced by ".", otherwise do not start with capital letter -> corrected to “.”
Line 197 "From the above figure" -> " From figure 2" -> Corrected
Line 247 "The following figures (See Figure 3 to Figure 4)" -> "Figure 3 and Figure 4" -> Corrected
Answer to comment 44: I accept your answer, but please give also a short explanation in the text (maybe it is already there, but I couldn't find it). -> Comment 44 was about the modulation information. The following phrase is added in the text L.180-182: “ For the ambiguity fixing step biases we do not use any additional bias for Galileo ambiguity fixing since a previous study [6] showed that the Galileo μ^s biases were compatible will all kind of receiver and modulations (such as L1C, L1W, L1X, ...)”
Reviewer 3 Report
L 187 - 189 and Figure caption 2: Mention in figure caption that the arbitrary biases were added for added Galileo-only and GPS+Galileo. It can be deleted from the text (L 187 - 189).
Figure captions 3 and 4: RMSs are in [m], although in the figure the units are in [mm].
L 272, 273, 277 and elsewhere: Please, mention somewhere how the improvements were calculated. Shouldn't it be the quadratic difference between two RMSs?
Author Response
Reviewer #3
L 187 - 189 and Figure caption 2: Mention in figure caption that the arbitrary biases were added for added Galileo-only and GPS+Galileo. It can be deleted from the text (L 187 - 189). -> Corrected, the sentence has been moved from the text to the caption.
Figure captions 3 and 4: RMSs are in [m], although in the figure the units are in [mm]. -> RMS values are changed to mm and captions are corrected to [mm]
L 272, 273, 277 and elsewhere: Please, mention somewhere how the improvements were calculated. Shouldn't it be the quadratic difference between two RMSs? -> Footnote added in the text, page 9 to explain how we calculate the percentages every time : « Here and elsewhere in the text, improvements in % are computed according to the formula:
where and are the values of before (hence reference) and after the change.”
Effectively, when speaking of RMS quadratic formulas can be used to compute the percentages but here, in this paper, the object is not a discussion on the statistical noise level of the different solutions and the percentage are indicated to give an idea of the RMS improvements as seen by the PPP/IPPP users.